# Effects of Exercise and Physical Activity Levels on Vaccination Efficacy: A Systematic Review and Meta-Analysis

**DOI:** 10.3390/vaccines10050769

**Published:** 2022-05-12

**Authors:** Petros C. Dinas, Yiannis Koutedakis, Leonidas G. Ioannou, George Metsios, George D. Kitas

**Affiliations:** 1FAME Laboratory, Department of Physical Education and Sport Science, University of Thessaly, 42100 Trikala, Greece; ioannoulg@gmail.com; 2Department of Physical Education and Sport Science, University of Thessaly, 42100 Trikala, Greece; y.koutedakis@gmail.com; 3Faculty of Education Health & Wellbeing, University of Wolverhampton, Walsall WS1 3BD, UK; g.metsios@uth.gr; 4Department of Nutrition and Dietetics, University of Thessaly, 42100 Trikala, Greece; 5Dudley Group NHS Foundation Trust and School of Sports and Exercise Science, University of Birmingham, Birmingham B15 2TT, UK; george.kitas@nhs.net

**Keywords:** vaccines and exercise, influenza, vaccines antibodies

## Abstract

Objective: We examined whether different intensities of exercise and/or physical activity (PA) levels affected and/or associated with vaccination efficacy. Methods: A systematic review and meta-analysis was conducted and registered with PROSPERO (CRD42021230108). The PubMed, EMBASE, Cochrane Library (trials), SportDiscus, and CINAHL databases were searched up to January 2022. Results: In total, 38 eligible studies were included. Chronic exercise increased influenza antibodies (standardized mean difference (SMD) = 0.49, confidence interval (CI) = 0.25–0.73, Z = 3.95, I^2^ = 90%, *p* < 0.01), which was mainly driven by aerobic exercise (SMD = 0.39, CI = 0.19–0.58, Z = 3.96, I^2^ = 77%, *p* < 0.01) as opposed to combined (aerobic + resistance; *p* = 0.07) or other exercise types (i.e., taiji and qigong, unspecified; *p* > 0.05). PA levels positively affected antibodies in response to influenza vaccination (SMD = 0.18, CI = 0.02–0.34, Z = 2.21, I^2^ = 76%, *p* = 0.03), which was mainly driven by high PA levels compared to moderate PA levels (Chi^2^ = 10.35, I^2^ = 90.3%, *p* < 0.01). Physically active individuals developed influenza antibodies in response to vaccination in >4 weeks (SMD = 0.64, CI = 0.30–0.98, Z = 3.72, I^2^ = 83%, *p* < 0.01) as opposed to <4 weeks (*p* > 0.05; Chi2 = 13.40, I^2^ = 92.5%, *p* < 0.01) post vaccination. Conclusion: Chronic aerobic exercise or high PA levels increased influenza antibodies in humans more than vaccinated individuals with no participation in exercise/PA. The evidence regarding the effects of exercise/PA levels on antibodies in response to vaccines other than influenza is extremely limited.

## 1. Introduction

Regular physical activity and exercise are prime modalities for the prevention of noncommunicable diseases [1,2] and have been advocated for resilience against infectious diseases (IDs) [3,4]. Aerobic training appears to improve cluster of differentiation 4 (CD4) function in human immunodeficiency virus (HIV) patients [5], while chronic exercise diminishes the harmful effects of obesity, aging, and chronic infections on T cells [6]. Similarly, individuals who consistently meet physical activity guidelines demonstrate a reduced risk for severe coronavirus disease 2019 (COVID-19) outcomes than those who are regularly physically inactive or partly active [7], while systematic, moderate-to-vigorous physical activity is associated with reduced risk of community-acquired ID and ID mortality [8]. Protection may, to some extent, be ascribed to the potential anti-inflammatory effects of regular exercise [9].

Vaccination is an established, simple, safe, and effective way of protecting people against ID [10]. Upon vaccination, regulatory T cells (Tregs) are produced, which differentiate further into specific cells to trigger cell-mediated immunity (CD8^+^) or antibody-mediated immunity (CD4^+^) [11]. The time course of the Treg response to vaccination depends on the presence of immunologic memory, which, if it exists, may activate Treg within 1–2 days [12], while the Treg-induced protection is variable and can be as short as six months, even though, in some cases (i.e., herpes zoster vaccine), this can be extended to three years [11]. Given that exercise may improve immune system through Treg subpopulation increases [13] and interleukin-10 levels, which affects tissue homeostasis by limiting host immune response to pathogens [14], it is logical to hypothesize that it can boost the immune responses to vaccination. Vaccination efficacy in relation to exercise/physical activity was investigated in a recent systematic review that examined the risk of community-acquired ID, improvements in immunization, and immunosurveillance in response to habitual physical activity [8]. However, there is no systematic review and synthesis of the quantitative evidence of the effect of different types/levels of exercise/physical activity on the efficacy of various vaccines in humans. Therefore, the purpose of this systematic review and meta-analysis was to examine whether different intensities of exercise and/or physical activity levels affected and/or associated with vaccination efficacy.

## 2. Methods

A systematic review and meta-analysis were conducted according to the Preferred Reporting Items for Systematic Reviews and Meta-analyses (PRISMA) guidelines [15] and registered with the International Prospective Register of Systematic Reviews (PROSPERO) database (registration number: CRD42021230108) [16].

### 2.1. Searching and Selection Processes

Two independent investigators (P. C. D. and L. I.) searched the PubMed, EMBASE, Cochrane Central Register of Controlled Trials, SportDiscus, and CINAHL databases up until January 2022. No restrictions were applied regarding the date of publication, participants’ health status, language of publication, or study design. The search algorithms are shown in the Appendix A. Reference lists of eligible publications were screened to identify studies that were not retrieved through the initial search. Three of the investigators (P. C. D., L. I., and Y. K.) selected eligible publications independently, and any disagreements were resolved through a referee investigator (G. D. K.).

### 2.2. Study Inclusion and Exclusion Criteria

Studies of any methodological design that involved human participants were included in the present study. Eligible studies combined: (a) an exercise/physical activity intervention and/or measurements of physical activity levels and/or comparison between physically active and non-physically active individuals along with (b) any type of vaccination that included measurements of relevant antibodies. As a control situation, we accepted studies that used either an appropriate control group (i.e., non-exercised/low physically active/physically inactive individuals) or baseline measurements that were compared with post intervention measurements. Animal studies, reviews, editorials, conference proceedings, magazines, and grey literature articles were excluded.

### 2.3. Study Quality Assessment

Y. K. and G. M. independently assessed the eligible studies for risk of bias, and any conflict was resolved through discussion with P. C. D. Even though we accepted studies with any methodological design, the selection process yielded only randomized controlled trials (RCTs), controlled trials without randomization (CTs), and cross-sectional studies (CSS). Hence, the updated Risk of Bias 2 (RoB2) Cochrane library [17], ROBINS-I [18], and Research Triangle Institute Item Bank (RTI-IB) [19] tools were used for RCTs, CTs, and CSS, respectively.

### 2.4. Data Extraction Strategy

Y. K. and G. M. extracted the data independently, with G. D. K. acting as referee in the case of disagreement. The following data were extracted: first author surname and date of publication; type of study (i.e., intervention, no intervention); methodological design (i.e., RCTs, CTs, or CSS); participants’ characteristics (i.e., age, gender, body mass index (BMI), and health and fitness status); interventions and/or comparisons of exercise/physical activity levels; type and time of vaccination; methods of evaluating vaccine efficacy; main results of the studies (Appendix A). The extracted data that were used in the meta-analyses and meta-regressions are available in an open depository [20,21].

### 2.5. Data Synthesis and Presentation

For eligible studies that did not provide numerical data to be used for a meta-analysis, a summarized narrative data synthesis was adopted. For studies suitable for meta-analysis, a random-effect model was used to account for heterogeneity due to the differences in study populations, type and time of vaccination, exercise/physical activity interventions, and study duration.

All meta-analyses were conducted using the RevMan 5.4.1, 2020, software (The Cochrane Collaboration, Oxford, UK) [22]. We used an inverse variance, continuous method to calculate antibody standardized mean differences (SMDs) as a result of any type of vaccination. To control individual differences in baseline values of antibodies, we calculated Δ scores (i.e., post intervention–baseline) for both the experimental (exercise intervention/physical activity measurements) and control groups at any follow-up time. Two studies [23,24], however, reported no baseline values and, as such, we used post intervention data. Standard deviations (*SD*s) for Δ scores were calculated based on current guidelines [25] as following:SD=((SD baseline^2+SD post^2)−(2∗corr∗SD baseline∗SD post)) 

Most, but not all, of the eligible studies provided nonparametric data. As nonparametric and parametric data cannot be mixed in a meta-analysis [25], we converted the means and *SD*s of parametric data into nonparametric data using well-established equations [26]: *mean*
$=ln\left(x\right)-\frac{1}{2}ln(\frac{S}{X}$
*+* 1); SD $=\sqrt{ln\left(\frac{S}{X}+1\right)\;}$ [26]. The decision to convert parametric data to nonparametric data was based on the fact that the majority of the eligible studies used nonparametric data, while the direction of the conversion (parametric to nonparametric and vice versa) does not affect the final outcome [26].

For 17 eligible studies [23,24,27,28,29,30,31,32,33,34,35,36,37,38,39,40,41], the means and *SD*s were depicted only in figures, as we were unable to retrieve data on the leading authors. Therefore, we used the WebPlotDigitizer 4.5, 2021, software (Rohatgi, USA) [42] to extract numerical data for our meta-analyses. The WebPlotDigitizer displays more than 5000 citations in Google Scholar and has previously shown intercoder reliability of 99.7% and agreement of 98.7% with the values reported in original research reports [43], and it has been suggested to be used in data synthesis of systematic reviews [44]. For one study [45], we calculated the means and *SD*s from medians and the 1st–3rd quartiles, according to: *mean* = (*q*1 + *m* + *q*3)/3; *SD* = (*q*3 − *q*1)/1.35 [25,46]. In addition, for eight studies [29,30,31,32,34,35,47,48], standard errors were converted into *SD*s using the equation: $\mathrm{SD}=standard\;error*\sqrt{n}$ [25]. For three eligible studies [24,49,50], we calculated the *SD*s from 95% confidence intervals (CIs) using the following equation in Excel: $SD=\sqrt{n}$ ∗ (*upper limit − lower limit*)/*TINV* (1-0.95; *N*1) [25]. Finally, for one eligible study [51] we calculated SD using the equation: *SD* = (2*SD* − *mean*)/2 [25,46].

“Acute exercise” refers to studies that used only one exercise session, “chronic exercise” refers to studies that used exercise programs for at least two weeks, while “physical activity levels” denotes studies that used measurements of physical activity in relation to levels of vaccination antibodies. The terms “<4 weeks” and “>4 weeks” represent antibody assessments in fewer or more than four weeks from vaccination, respectively. Even though the kinetics of antibody response to vaccination are extremely complex [52], we chose the 4 week threshold, because by the 4th week post vaccination, many data suggest that individuals who are likely to respond to vaccinations have already responded [53,54,55]. Since different scales were adopted in the eligible studies, we used SMDs instead of absolute mean differences to standardize the findings to a uniform scale [25]. We calculated the SMDs of antibodies for the following studies: (a) influenza vaccine antibodies of RCTs and CTs for acute exercise, (b) pneumococcal vaccine antibodies of RCTs and CTs for acute exercise, (c) influenza vaccine antibodies of RCTs and CTs for chronic exercise, (d) pneumococcal vaccine antibodies of RCTs and CTs for chronic exercise, (e) influenza vaccine antibodies of CSS that measured physical activity levels, and (f) pneumococcal vaccine antibodies of CSS that measured physical activity levels. Even though we retrieved studies for vaccine antibodies of the human papillomavirus, diphtheria, tetanus, meningococcal, and coronavirus, these appeared only once for each category and, therefore, no meta-analysis was conducted [25]. However, we included these studies in the narrative data synthesis. Where pertinent, we used subgroup meta-analyses to calculate SMD with respect to exercise type (aerobic vs. resistance vs. combined (aerobic + resistance)), exercise intensity, type of vaccine antibodies and level of physical activity and age (young vs. old). We considered heterogeneity as significant if *p* < 0.10, while we interpreted the I^2^ index based on standard guidelines [25]. Publication bias was assessed using funnel plots but only for those meta-analyses that included >10 studies/entries [25]. Finally, we conducted meta-regression analyses to test associations between antibodies following vaccination and exercise/physical activity while taking into account the Δ scores of the following moderators: mean BMI, mean age, and percentage of each gender for each study. For the meta-regression analyses, the “metafor” package in the R language (Rstudio, version 1.3.1093, PBC, Boston, MA, USA) was used in a mixed-effect model using the SMD [25,56,57].

### 2.6. Evidence of Effectiveness

We evaluated the quality of evidence of each meta-analysis via the grading of recommendations assessment, development, and evaluation (GRADE) analysis [25,58].

## 3. Results

Reporting information is shown in the relevant PRISMA checklist (Appendix A).

### 3.1. Results of Searching and Selection Processes

Of the 2589 retrieved publications, 835 were duplicates while 1666 did not meet the prespecified eligibility criteria and were excluded by title and abstract [25]. Fifty-three publications that did not fulfill the inclusion criteria were also excluded. Thirty-eight studies were included in the systematic review (i.e., 35 classified as eligible and 3 that were found in the reference lists of the eligible studies); see the PRISMA flow diagram (Appendix A). A full list of the excluded publications can be found in the Appendix A.

### 3.2. Characteristics of the Included Studies and Risk of Bias Assessment Outcomes

The studies that were included in this systematic review were published between 1996 and 2021 and involved 5984 healthy participants, 898 participants with autoimmune rheumatic disease, and 137 patients with coronary artery disease. Seventeen studies were RCTs, five were CTs, and 16 were CSS. Four RCTs [31,34,35,59] and one CT [45] examined the effects of acute aerobic exercise, while seven RCTs [28,30,38,49,50,60,61] studied acute resistance exercise. Four RCTs [47,48,62,63] examined the effects of chronic aerobic exercise, and one RCT [64] investigated chronic physical activity levels. One CT [24] studied the effects of chronic involvement in taiji and qigong, one RCT [41], two CTs [27,39] the effects of chronic combined exercise (aerobic + resistance), and one CT [36] the effects of a nonspecific chronic exercise program. Finally, 16 CSS [23,29,32,33,37,40,51,65,66,67,68,69,70,71,72,73] examined the relationship between physical activity levels and antibodies in response to various vaccines.

Most studies (*n* = 29) [23,24,27,28,29,30,31,32,33,34,35,36,37,38,39,41,47,48,49,61,62,63,65,66,67,68,69,70,72] utilized influenza vaccines, six studies [34,40,45,50,51,64] pneumococcal, two SARS-CoV-2 [71,73], two tetanus [32,45], one human papillomavirus [60], one diphtheria [45], and one meningococcal vaccines [59]. The risk of bias assessment outcomes are shown in the Appendix A.

### 3.3. Narrative data Synthesis Results

Twelve studies were included in the narrative data synthesis. The outcomes appear in Table 1.

Two studies examined the effects of acute aerobic exercise and fitness level on antibodies in response to tetanus vaccination and found either no effect [32] or an adverse effect of exercise (i.e., exercise disrupted development of vaccine antibodies) [45]. Two studies detected that physical activity levels were positively associated with antibodies in response to SARS-CoV-2 vaccination [71,73]. For influenza vaccinations, three studies found a positive association of physical activity levels with antibody development [66,67,69], and one revealed no association [70]. Moreover, one study showed no effect of routine daily exercise on development of antibodies in response to influenza vaccinations [72]. In addition, acute exercise had no effect on antibodies in response to human papillomavirus vaccination [60], while it had an adverse effect (i.e., exercise disrupted development of vaccine antibodies) on diphtheria vaccination [45] and a positive effect on meningococcal vaccination but only in men [59]. Finally, physical activity had no effect on antibodies in response to pneumococcal vaccination [64]. All eligible studies that examined the effects of acute exercise on vaccination efficacy reported that the vaccinations were administered after the exercise session.

### 3.4. Acute (One Session) Exercise: Meta-Analysis Outcomes from RCTs and CTs

We found no effect of acute exercise on antibodies in response to influenza vaccination (*p* > 0.05). Subgroup analysis did not reveal statistically significant SMDs with respect to exercise type (aerobic vs. resistance; *p* > 0.05, Appendix A), vaccine antibodies type (H1N1 vs. H3N2 vs. B; *p* > 0.05, Appendix A), exercise intensity (low (<65%) heart rate maximum (HRmax), one repetition maximum (1RM) vs. high (>75%) HRmax, 1RM; *p* > 0.05, Appendix A) [74,75], age (<50 years vs. >50 years; *p* > 0.05, Appendix A) as well as time of vaccine antibody measurements (<4 weeks vs. >4 weeks; *p* > 0.05, Appendix A). Regarding pneumococcal vaccinations, acute exercise had no effect on antibody development (*p* > 0.05). Subgroup analysis showed no statistically significant SMD with respect to exercise type (aerobic vs. resistance; *p* > 0.05, Appendix A), exercise intensity (low (<65%) HRmax, 1RM vs. moderate (65–75%) HRmax, 1RM vs. high (>75%) HRmax, 1RM; *p* > 0.05, Appendix A), age (<31 years vs. 57 years; *p* > 0.05, Appendix A), while all measurements for pneumococcal antibodies were performed in <4 weeks from vaccination and, as such, no subgroup analysis for the time of measurements was performed.

All the eligible studies that examined the effects of acute exercise on vaccination efficacy, reported that the vaccinations were administered after the exercise session. Furthermore, 10 eligible studies [28,30,31,34,35,38,45,49,59,60] that examined the effects of acute exercise on vaccination efficacy reported no data, while two studies reported no severe immediate symptomatic responses due to the administration of vaccines [61,72]. Three studies [31,35,50] reported that interleukin 6 (IL-6) did not affect vaccination efficacy. Therefore, no firm conclusions can be drawn as to whether site or symptom vaccine reactions and inflammatory responses played a role in vaccination efficacy.

### 3.5. Chronic Exercise (Duration > 2 Weeks): Meta-Analysis Outcomes from RCTs and CTs

We detected an effect of chronic exercise on influenza antibodies (SMD = 0.49, CI = 0.25–0.73, Z = 3.95, I^2^ = 90%, *p* < 0.01; Figure 1 and Appendix A).

Subgroup analysis revealed that this effect was mainly driven by aerobic exercise (SMD = 0.39, CI = 0.19–0.58, Z = 3.96, I^2^ = 77%, *p* < 0.01; Appendix A) as opposed to combined (aerobic + resistance; *p* = 0.07) and other exercise types (i.e., Taiji and Qigong, unspecified; *p* > 0.05). However, there was not a statistically significant SMD between subgroups in exercise types (*p* > 0.05; Appendix A). Furthermore, no statistically significant SMDs were detected neither between low (<65% HRmax, 1RM) and moderate (65–75% HRmax, 1RM) exercise intensities (*p* > 0.05; Appendix A) nor between individuals of <59 years old and >59 years old (*p* > 0.05; Appendix A) in the effect of chronic exercise on influenza vaccine antibodies. Similarly, there were no statistically significant SMDs between influenza antibodies measured <4 and >4 weeks post vaccination in response to chronic exercise (*p* > 0.05; Appendix A). We detected, however, a statistically significant SMD between types of influenza vaccine antibodies (H1N1 vs. H3N2 vs. B vs. all strains) in response to chronic exercise (Chi^2^ = 11.23, I^2^ = 73.3%, *p* = 0.01; Appendix A). In this regard, H1N1 influenza antibodies were increased more than H3N2 and B types, while the B type was not affected (*p* > 0.05), in response to chronic exercise (Appendix A). Finally, within the aerobic exercise group, we also found that H1N1 influenza antibodies increased more than H3N2 and B types, while the B type was not affected (*p* > 0.05), in response to chronic exercise (Chi^2^ = 10.11, I^2^ = 80.2%, *p* < 0.01; Appendix A). Meta-regression analysis revealed no associations between antibodies following influenza vaccination and chronic exercise and mean BMI, mean age, and the percentage of males/females who participated in each study (*p* > 0.05).

### 3.6. Physical Activity: Meta-Analysis Outcomes from CSS

We detected no statistically significant effects of physical activity levels on pneumococcal antibodies in response to vaccination (*p* > 0.05; Appendix A). Our data, however, revealed that physical activity levels positively affected antibodies in response to influenza vaccination (SMD = 0.18, CI = 0.02–0.34, Z = 2.21, I^2^ = 76%, *p* = 0.03; Appendix A). Subgroup analysis revealed that high physical activity levels (i.e., >50 km running/week or >20 min vigorous exercise three times/week or >10,924 kj/60 kg/day or >16,640 steps/day or a fitness level of >44 mL/kg/minute of maximum oxygen consumption (VO_2_max)) positively affect antibodies in response to influenza vaccination (SMD = 0.38, CI = 0.15–0.60, Z = 3.31, I^2^ = 78%, *p* < 0.01; Figure 2 and Appendix A) as opposed to moderate physical activity levels (i.e., <6 km running/week or <20 min vigorous exercise three times/week or between 8823–10,924 kj/60 kg/day or <9050 steps/day or a fitness level of <44 mL/kg/minute of VO_2_max; Chi^2^ = 10.35, I^2^ = 90.3%, *p* < 0.01; Figure 2 and Appendix A).

We also found that physically active individuals developed influenza antibodies in response to vaccination in >4 weeks (SMD = 0.64, CI = 0.30–0.98, Z = 3.72, I^2^ = 83%, *p* < 0.01) as opposed to <4 weeks (*p* > 0.05; Chi^2^ = 13.40, I^2^ = 92.5%, *p* = 0.0003; Appendix A) post vaccination. Subgroup analysis also revealed that physically active old individuals (64–75 years old) developed more influenza antibodies in response to vaccination than physically active young individuals (21–23 years old; Chi^2^ = 26.25, I^2^ = 96.2%, *p* < 0.01; Figure 3 and Appendix A). In this regard, in young individuals, influenza vaccine antibodies’ development appeared to be disrupted in response to physical activity levels (SMD = −0.17, CI = −0.33–0, Z = 2.00, I^2^ = 61%, *p* = 0.05; Figure 3 and Appendix A). However, the later analysis also included individuals with moderate physical activity levels, which already showed no effect of physical activity levels on influenza vaccine antibody development (*p* > 0.05, Figure 2). For this reason, we performed a subgroup analysis for individuals with high physical activity levels only, which also revealed that physically active older individuals (64–75 years old) developed more influenza antibodies in response to vaccination than physically active young individuals (21–23 years old; Chi^2^ = 21.04, I^2^ = 95.2%, *p* < 0.01; Appendix A) but with no adverse effect (i.e., disrupted antibody development due to the physical activity) of physical activity on influenza vaccine antibody development of young individuals (21–23 years old; Appendix A). Regarding different influenza antibodies vaccine types, we found that physical activity positively affected mainly the influenza antibodies of the H3N2 and B vaccine types (Chi^2^ = 15.39, I^2^ = 80.5%, *p* < 0.01; Appendix A).

The meta-regression analysis revealed a positive association between influenza vaccine antibody development due to the physical activity in relation to age (*R*^2^ = 31.23%, I^2^ = 75.32%, *p* < 0.01, Appendix A), while influenza vaccine antibody development was associated with a higher percentage of physically active men and with a lower percentage of physically active women (*R*^2^ = 8.42%, I^2^ = 80.24%, *p* = 0.04, Appendix A).

Regarding immediate symptomatic responses due to the administration of vaccines, 21 of the eligible studies that examined the effects of chronic exercise on vaccination efficacy and the associations of physical activity levels with vaccination efficacy reported no data [23,24,27,29,32,33,36,37,39,40,41,47,48,51,63,64,67,68,69,70,71]; one study reported no adverse effects [66], one study reported seven minor events [62], and one study reported 98.1% of local and 59.9% of systemic adverse effects in the 1st dose as well as 97.8% of local and 90.3% of systemic adverse effects in the 2nd dose [73]. Furthermore, out of the 24 eligible studies that examined the effects of chronic exercise on vaccination efficacy and the associations of physical activity levels with vaccination efficacy, two studies reported no changes in IL-10 [39,48], and one study reported reduced IL-10 after the intervention, which was associated with reduced cell immune responses [32]. Moreover, two studies reported no changes in IL-6 due to the intervention [39,69], and two studies reported no changes in C-reactive protein [37,73]. These outcomes cannot form a clear picture of whether side effects and/or inflammatory responses played a role in vaccine responsiveness.

### 3.7. GRADE Analysis Outcomes

Our GRADE analysis outcomes appear in Table 2, while the detailed evaluation of its components can be found in Appendix A. The meta-analysis outcomes of the effects of chronic exercise of all types and chronic aerobic exercise on influenza vaccine antibodies showed a moderate quality of evidence, while the meta-analysis outcomes of the effects of physical activity and high physical activity levels on influenza vaccine antibodies showed a low quality of evidence.

## 4. Discussion

The aim of the current systematic review and meta-analysis was to examine whether different intensities of exercise and/or physical activity levels affected and/or associated with efficacy of vaccination.

### 4.1. Summary of Main Findings

The meta-analysis outcomes showed that chronic aerobic exercise increased influenza vaccine antibodies compared to non-exercise, while it had no effect on pneumococcal antibodies. It was also reported, by only one study that offered no data to be included in the meta-analysis, that chronic exercise increased influenza vaccine antibodies [66]. The meta-analysis outcomes also demonstrated that high physical activity levels increased influenza vaccine antibodies as opposed to moderate and/or low physical activity levels as well as to no physical activity. This effect appeared to occur in >4 weeks from the time of vaccination, and it was evident in older (i.e., >64 years old) compared to younger individuals (i.e., <23 years old). We also detected that the percentage of male gender was positively associated with influenza vaccine antibody development in response to physical activity, while the percentage of female gender was negatively associated with physical activity. In addition, two studies [67,69], which did not offer data to be included in the meta-analysis, reported a positive effect of physical activity levels on influenza vaccine antibody development. Two further studies that also offered no data to be included in the meta-analysis stated positive effects of physical activity levels on SARS-CoV-2 vaccine antibody development [71,73].

The present data have shown that acute exercise had no effect on influenza, pneumococcal and human papillomavirus vaccine antibodies. However, one study [64] revealed that acute exercise increased meningococcal vaccine antibodies in men, while another reported an adverse effect of acute exercise on diphtheria and tetanus vaccine antibodies [45].

### 4.2. Completeness and Applicability of Evidence

There was sufficient evidence to assess the effects of exercise/physical activity on influenza (28/35 included studies—80%) and, to a lesser extent, pneumococcal (7/35 studies—20%) vaccine efficacy. However, the evidence for other vaccines (SARS-CoV-2 (2/35), tetanus (2/35), meningococcal (1/35), diphtheria (1/35), and human papillomavirus (1/35 study)) was extremely limited.

Regarding acute exercise, one eligible study reported adverse effects of an exercise session in response to tetanus and diphtheria vaccines [45]. This phenomenon was attributed to the fact that the time of the antibody measurements (48 h post vaccination) did not allow for full antibody development [45]. A similar study also reported no difference in B-cell function between intense exercisers and controls, although 15 days later, tetanus toxoid vaccine antibody titers were higher in the exercisers than the controls [76].

We found a moderate quality of evidence that chronic aerobic exercise increased influenza vaccine efficacy, even in individuals participating in low-intensity exercise. We also found that high physical activity levels improved influenza vaccine efficacy; however, the quality of the evidence retrieved for this analysis was found to be low. Involvement in physical activity was also positively associated with higher antibody titers in response to influenza vaccines in most of the studies narratively analyzed herein. These findings concur with available data demonstrating a beneficial effect of chronic exercise on the immune system through Treg increases [13]. Interestingly, high physical activity levels—usually displayed by chronic exercisers—not only increased Treg but also interleukin-10 levels, which affects tissue homeostasis by limiting host immune response to pathogens [14]. High fitness levels—usually displayed by chronic exercisers—associated with increased Tregs in CD4^+^, CD25^high^, and CD127^low^ and memory of Tregs in CD4^+^, CD25^+^, and CD39^+^, even in obesity [77]. Chronic exercise may also increase CD3^+^, CD4^+^, and CD8^+^ T-cell counts in the elderly, [78] while Tregs are significantly higher in athletes than in nonathletes [79,80]. These outcomes indicate that chronic exercise may trigger thymic maintenance by disrupting thymosuppressive factors, such as leukemia inhibitory factor, oncostatin M, and IL-6, which elevate with age [79]. Even though strenuous exercise bouts may decrease Th1 cells (lineage of CD4+ effector T cell), these usually return to normal levels within 24 h [80]. However, the repeated exercise bouts (i.e., chronic exercise) causes more sustained changes to these immune parameters, which may create high levels of anti-inflammatory immune cells [80]. Therefore, chronic involvement in exercise and/or high physical activity levels may positively affect immune responses that may be linked to vaccination efficacy.

We found positive associations (*R*^2^ = 31.23, I^2^ = 75.32%) between age and the SMD of antibodies of physical activity levels in response to influenza vaccines, indicating that older individuals may produce higher antibody levels than their younger counterparts. Although this finding may seem to refute the phenomenon of age-related immunosenescence [81], there are precedents for vaccines working just as well in older as in younger adults [82]. In fact, a recent RCT revealed that in a cohort of participants >65 years, vaccine efficacy was highest in the older ones [83], while a similar efficacy for COVID-19 vaccines was found with regard to age in a cohort of >36,000 participants 16 to >75 years old [84]. It is also possible that older adults who are chronic exercisers may overcome, to a certain extent, immunosenescence, since exercise associated with a decrease in senescent T lymphocytes [85]. However, findings indicating that old individuals may produce higher influenza vaccine antibodies than younger individuals in response to physical activity should be considered for future research.

We detected positive associations (*R*^2^ = 8.42, I^2^ = 80.24%) of the SMD of antibodies of physical activity minus control in response to various vaccines and percentage of males, while we found negative associations (*R*^2^ = 8.42, I^2^ = 80.24%) for the same parameters in females. The reasons for this biological difference are unclear and may include genetic, hormonal, and environmental factors, or their combination. Antibody responses to bacterial and viral vaccines are thought to be usually higher in females than males [86,87], although it is notable that when a total of 43,548 participants underwent randomization for vaccination with the Pfizer/Biontech COVID-19 vaccine, BNT162b2 (21,728 with placebo), similar vaccine efficacy was observed in subgroups defined by age, gender, race, ethnicity, baseline BMI, and the presence of coexisting conditions [84]. However, none of these studies considered the effects of exercise/physical activity, and when the impact of gender on the immune response to exercise/physical activity was examined by some studies, a rather different picture emerges. For instance, both the neutrophil and lymphocyte responses to a swimming session were weaker in females than males [88]. Likewise, it has been found that the effect of a maximal incremental swimming task on immunity is gender dependent and more noticeable in men [89]. This is yet another area that merits further investigation.

### 4.3. Strengths and Potential Biases in the Review Process and Disagreement with Previous Systematic Reviews

The strengths of our systematic review include: (a) the use of suitable algorithms with standardized indexing terms in the search procedure, which may have retrieved publications that used alternative keywords to describe the same concept [25]; (b) we utilized robust searching and screening procedures, risk of bias assessment, and data extraction, and did not exclude studies based on language and time of publication; (c) the quality of our meta-analyses was evaluated through the GRADE analysis.

The present review also has certain limitations, which include: (a) limited ability to fully examine the effects of exercise/physical activity on antibody responses to vaccines other than influenza; (b) inability to test the dose–response of antibodies to exercise response in vaccines other than influenza; (c) no inclusion of grey literature or search of websites of relevant scientific organizations.

Technological advances in measuring antibodies over the period that the included studies in the systematic review were conducted (1996–2021) may have affected the current outcomes. We found no or limited data in the eligible studies, to examine whether inflammatory markers or catecholamines played role in vaccination efficacy. In addition, the reporting information in the eligible studies on the immediate symptomatic responses due to administration of vaccines was also extremely limited. Generally, our systematic review revealed, apart from influenza vaccinations, a lack of well-powered studies that considered (a) metrics associated with physical fitness and acute exercise and (b) metrics associated with the equally complex immune response. We also acknowledge the heterogeneity of the available evidence regarding populations, types and times of vaccinations, exercise/physical activity interventions, and study durations. To comply with a wider adoption of evidence synthesis [90], we used a random-effect model meta-analysis, which may ignore heterogeneity [25]. This allowed us to form a meaningful conclusion, while heterogeneity was considered in the GRADE analysis, the quality of the evidence was reduced, where pertinent.

A recent systematic review [8] examined the risk of community-acquired ID, improvements in immunization, and immunosurveillance in response to habitual physical activity. This aim, however, was different from the current systematic review, as we examined the effects of acute and chronic structured exercise as well as participation in physical activity on the antibodies of various vaccines. In addition, unlike the aforementioned study, we included all available methodological designs—although the selection process yielded only RCTs, CTs, and CSS—for eligible studies, and we did not exclude studies based on the participants’ physical fitness levels, according to recent recommendations [90].

### 4.4. Statement on the Significant Deviations Methods from the Published Protocol

We report no deviations from the published protocol [16].

## 5. Conclusions

Healthy individuals involved in chronic aerobic exercise increased influenza antibodies in response to vaccination more than their counterparts who did not participate in such exercise. This beneficial effect of chronic exercise was positively associated with the male gender and increasing age. High physical activity levels increased antibodies in healthy individuals in response to influenza vaccination, and this effect appeared >4 weeks from the time of vaccination; however, this outcome should be treated with caution due to the low quality of the evidence. Acute exercise (i.e., one session) had no effect on antibodies in response to influenza vaccination. Evidence regarding the effects of exercise/physical activity levels on antibodies in response to vaccines other than influenza is extremely limited and, therefore, no firm conclusions can be drawn. As vaccination is a proven means for combating ID, including COVID-19, it is crucial to contemplate the current results in an attempt to improve vaccination efficacy. Future RCTs should examine the effects of acute and chronic exercise as well as high and moderate physical activity levels on antibodies of various vaccines, especially human papillomavirus, tetanus, and zoster, where the evidence is either extremely limited or missing. Future studies should also examine whether the health status of individuals who undertake vaccination plays a role in vaccination efficacy in relation to exercise and physical activity levels. In this light, inflammatory markers or catecholamines that play a role in vaccination efficacy should also be examined in future studies.

## Figures and Tables

**Figure 1 vaccines-10-00769-f001:**
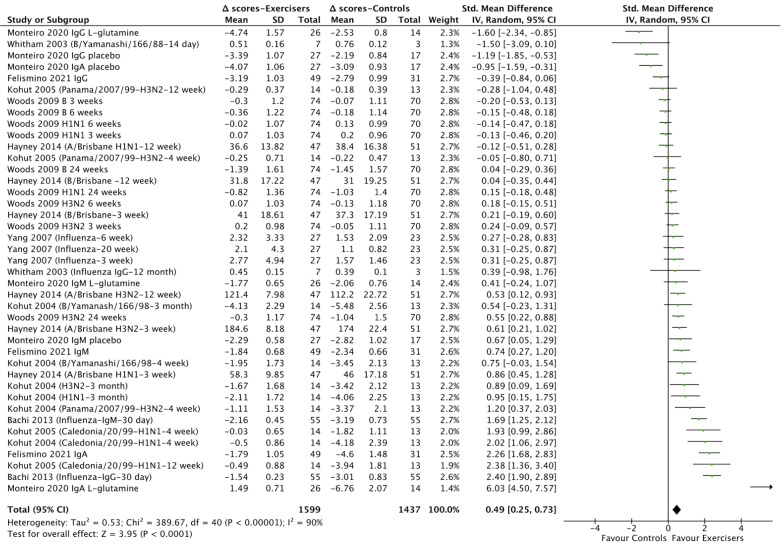
Forest plot of the effect of chronic exercise on influenza vaccine antibodies. Δ scores: post intervention–baseline; SD: standard deviation; 95% CI: 95% confidence interval.

**Figure 2 vaccines-10-00769-f002:**
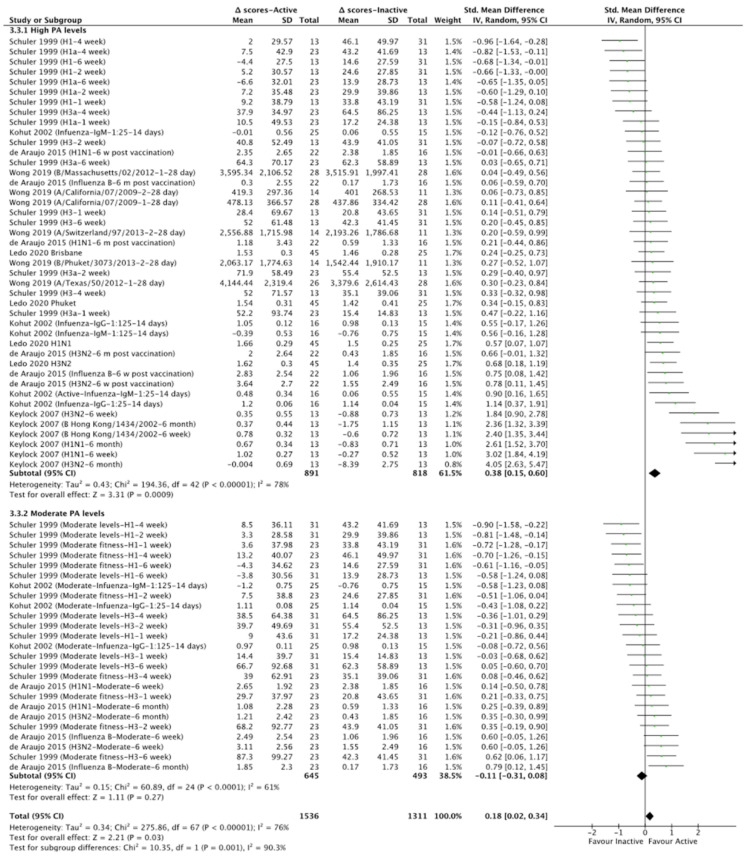
Forest plot of the effect of physical activity levels on influenza vaccine antibodies (subgroup analysis for high and moderate physical activity levels). Δ scores: post intervention–baseline; SD: standard deviation; 95% CI: 95% confidence interval.

**Figure 3 vaccines-10-00769-f003:**
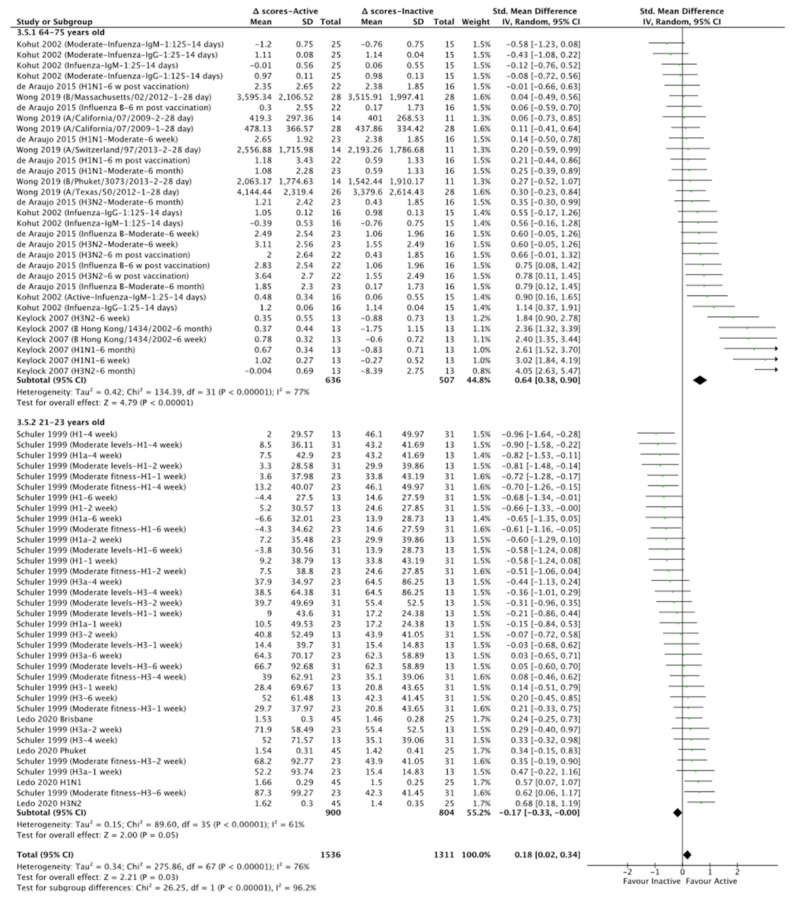
Forest plot of the effect of physical activity levels on influenza vaccine antibodies (subgroup analysis for age). Δ scores: post intervention–baseline; SD: standard deviation; 95% CI: 95% confidence interval.

**Table 1 vaccines-10-00769-t001:** Narrative data synthesis outcomes for the eligible studies that were not included in a meta-analysis.

Study	Type of Intervention	Type of Vaccination	Effects on Antibodies or Association with Physical Activity Levels-Time of Antibody Measurements
Bruunsgaard 1997	Acute aerobic exercise	Tetanus	Adverse effect of exercise: 48 h post vaccination
Bohn-Goldbaum 2019	Acute resistance exercise	Human papillomavirus	No effect: 7.5 months post vaccination
Bruunsgaard 1997	Acute aerobic exercise	Diphtheria	Adverse effect of exercise: 48 h post vaccination
Edwards 2008	Acute aerobic exercise	Meningococcal	Men showed a positive effect of exercise. No effect for women: 4 and 20 weeks post vaccination
Kenzaka 2021	Acute routine daily exercise	Influenza	No effect on vaccination day
Keylock 2007	Fitness levels	Tetanus	No effect: 6 weeks and 6 months post vaccination
Keshtkar-Jahromi 2010	Chronic exercise	Influenza	Antibodies were positively associated with regular exercise: 1 month after vaccination
Gualano 2021	Physical activity levels	SARS-CoV-2	Physical activity enhanced SARS-CoV-2 vaccine immunogenicity: not reported
Mitsunaga 2021	Physical activity levels	SARS-CoV-2	Lack of outdoor exercise was a suppressor of antibody responses: 7–20 days after vaccination
Schuler 2003	Physical activity levels	Influenza	Positive association of H3N2 antibodies with physical activity levels in the 1st week post vaccination: 1, 2, 4, and 6 weeks post vaccination
Segerstrom 2012	Physical activity levels	Influenza	Above average physical activity was associated with higher antibody response: 2–4 weeks post vaccination
Stewart 2018	Physical activity levels	Influenza	No association: 4 weeks post vaccination
Long 2013	Physical activity levels	Pneumococcal	No effect: 4 weeks and 6 months post vaccination

**Table 2 vaccines-10-00769-t002:** GRADE analysis outcomes.

Outcome on Influenza Vaccine Antibodies	Relative Effect SMD (95% CI)	Number of Participants	Certainty of Evidence (GRADE)
Chronic exercise (all types)	0.49 (0.25–0.73)	3036	Moderate ⨁⨁⨁◯
Chronic aerobic exercise	0.37 (0.18–0.56)	2174	Moderate ⨁⨁⨁◯
Physical activity levels	0.18 (0.02–0.34)	2847	Low ⨁⨁◯◯
High physical activity levels	0.53 (0.29–0.78)	1357	Low ⨁⨁◯◯

CI: confidence interval; SMD: standardized mean difference; ⨁⨁⨁◯: Moderate; ⨁⨁◯◯: Low.

## Data Availability

Extracted data used in the meta-analyses and meta-regressions were deposited in the figshare database. Link: https://figshare.com/s/ed1e3ab7958d07fd7fe0, https://figshare.com/s/0f66e4d004b6e889ae5d (accessed on 16 March 2022).

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
