# Peer review of "Effects of Exercise and Physical Activity Levels on Vaccination Efficacy: A Systematic Review and Meta-Analysis"

_vaccines, 2022, doi:10.3390/vaccines10050769_

Round 1

Reviewer 1 Report

The authors have provided a systematic review via meta-analysis of clinical studies that have addressed the effects of exercise or physical activity on immunity derived from vaccination in multiple infectious disease settings. The no doubt rigorous review of these studies and subsequent analysis of the work took a great deal of time to address. At the conclusion of the inclusion/exclusion process, the 1600+ studies across several study databases were filtered down to just 38. The effects of exercise on musculoskeletal tissue is well-known as are the benefits of being active on the immune system. There are many different modalities of exercise and many forms of physical activity one can engage in, therefore, there is importance in the aims of this manuscript. What is less obvious is the impact exercise on the up- or down-regulation of antibody production immediately following vaccination for various infectious diseases.  The authors make great use of available data to start understanding these differences, if they exist, and in which direction they are drive due to the nature of the exercise or physical activity performed. There are some points below that may help clarify the authors’ points and contribute to the clarity of the manuscript. 

Abstract and Introduction 

Both sections are well-composed.  More information could be provided in the introduction which could serve as a background to cellular responses to vaccines and the time course of antibody generation. 

It was unclear why the window of efficacy was either less than or greater than 4w. (lines 29-30) 

Methods 

Line 68: Should participant’s health status be a major consideration in these studies? 

Lines 79-80: Please explain the absence of exclusion criteria for the controls. 

Line 101: What does ”not provide suitable data” mean in terms of study eligibility? 

Line 106: Provide company of RevMan 5.4 software. 

Line 123: Provide company of WebPlotDigitizer. 

Liune 147-149: Are the other factors taken into account at all…starting patient weight, activity levels,  

Results 

The authors report positive effects in chronic exercisers vaccinated with influenza and that this is a byproduct of aerobic exercise.  

Line 164: was does excluded by title mean?...what were the criteria for the exclusions? 

Line 165: “not fulfilling” is a typo. 

Line 166: Why were there so many studies excluded from the analysis….seemingly 1600+ manuscripts/studies that were whittled down to 38…that’s quite dramatic.  Do the authors believe that opening their criteria more, they would’ve arrived at stronger outcomes? 

Line 188: Typo between “studies” and “included”. 

Why was strength training not included in the analysis? Please include an explanation in the limitations of the discussion. 

Discussion 

Line 308: maybe more appropriate to say different intensities of exercise. Also, perhaps important to mention that these are reported findings and not necessarily all inclusive as the cohorts studied and the exercise regimens incomplete.  

I would begin Section 4.1 discussing the positive outcomes of the findings that support the highlights made in the abstract and conclusion, that chronic exercise increased influenza vaccine antibodies in men.  

Are the following effects ever explained as possibilities: 1. Adverse effects of acute exercise in diptheria and tatnus? 2. Positive effects of high physical activity in influenza vaccine antibodies in 64+ men versus those that are young?  

Line 331: Completeness is misspelled.  

Section 4.2: it would be beneficial for the reader if the authors explained the implications of these upregulations in relation to their anti-viral activity and downstream immunogenicity. 

Conclusions 

Reword the first sentence. 

Nothing is mentioned in the conclusion regarding acute exercise, nor any concluding remarks about the other infectious diseases tested. 

Table 1 

Are there differences as to the time the antibodies were queried after the interventions? If so, can a statistical measured be shown to indicate that these differences (if there were) are significant and thus would lead to different outcomes? 

Perhaps rearrange Table 1 to reflect the findings in a common format.  

Consider adding a lengthier legend. 

Table 2 

Trim some of the repetitive words from the columns as they detract from the message.  

Figure 1-3 

Quality of the text in the figures need to be improved. 

Author Response

Dear Reviewer,

We wish to thank you for your effort in reviewing the manuscript and for your constructive and helpful comments. We understand that the short comings identified during the review process were important and we appreciate the fact that you valued the interest of our paper and have given us a chance to revise it. We made appropriate changes in the paper based on the comments that we received, and we believe that the revisions have markedly improved the quality of our manuscript.

The appropriate responses to all points that you raised appear in the uploaded word file, in bold font. Red font is used to indicate revised parts of the text. The changes/modifications in the revised paper appear as track changes. The line numbers appearing in our responses to your comments are based on the copy of the revised manuscript that includes track changes.

Reviewer 2 Report

This is a very interesting and well done review of a particularly relevant and important topic. 

General Comments: I would suggest to the authors that what is really shown here is a lack of well-powered studies that take into account the expertise needed in the two families of variables: 1) metrics associated with physical fitness and acute exercise; and 2) metrics associated with the equally complex "immune response." Only with influenza did the previous literature suggest a beneficial relationship between some physical activity parameter, but that might be because there were more good studies found for influenza than for the other vaccines. For example, a single study on HPV using a single resistance acute exercise bout is hardly enough to draw any conclusions. When one reads the original gardisil company docs, buried in there is the suggestion that obesity in adolescents (related, of course, to physical fitness) impaired antibody responses.

Specific comments:

  1. page 3, need for WebPlot digitizer--another reason to promote accessibility for the actual data in all pubs. Just like the genomic studies have done for many years.
  2. page 4: "fulfilled" not "fulfilling"
  3. page 4: 1996 to 2021. That is a long time and the authors might comment on whether possible improvements in techniques to measure antibodies have changed over that interval. We know they have, we know a heck of a lot more now than then,, for example, about Tregs, a mechanism suggested for PA vaccine benefit.
  4. page 5: OK, acute exercise. We know the innate and adaptive immune response to an acute exercise response is dramatic but short-lived. It is true that one of the main things that the profound perturbation of brief intense exercise is to mobilize gobs of leukocytes, primarily from the lungs, but marginal pools attached to the vascular may also be involves as well as some mobilization from marrow. So when the vaccine is administered after an acute exercise bout is critical to any likely impact on vaccine responsiveness. Did any study administer the vaccine immediately prior to an acute exercise bout? Did any study look at the immediate symptomatic response to the vaccine in light of acute (or chronic) exercise (e.g., fever, vaccine site pain, etc.)? Other immune and inflammatory responses to acute exercise that could have an effect on vaccine responses (e.g., sudden increase in catecholamines, IL-6, and other cytokines) could play a role. Finally, any way that there might be an approach to correlate in the studies the actual amount of acute exercise (e.g., watts, percent VO2max, duration; at least for the aerobic exericse studies).
  5. "Chronic Exercise." Again, it would have been interesting to determine in the training studies whether or not there was a "training effect?" Did any of the studies report on changes in VO2max, muscle mass, or body composition. There is an emerging concept of normal weight obesity in which the inflammatory effect of adiposity is gauged not just by whether someone is obese, but by the ratio of fat to lean tissue. Even normal weight adolescents, for example, demonstrated a relationship between percent body fat and IL-6 and CRP.
  6. page 7. Physical activity levels. THe authors realize that self report PA is not very good. Did any of the studies use accelerometers or heart rate monitors to assess habitual physical activity?
  7. Conclusions: My reading of this is as follows: the fact that the systematic review found beneficial effects for any vaccine (influenza) is remarkable. Most of the studies reviewed were not any good. A take home message, I think, would be to spur interest in actually conducting appropriately designed clinical trials. Obvious targets could be HPV in adolescents (mostly immune naive subjects); zoster in the older individuals; and tetanus (cause everybody gets the vaccine), but the prior exposure might be challenging. Although, there is some literature on negative effects of obesity in kids on tetanus antibodies.

Author Response

(The authors gave the same response as above.)

Round 2

Reviewer 2 Report

The authors did an excellent job in responding to the critique.